# DISTRIBUTION-GUIDED EXPERT ROUTING FOR IMBALANCED MOLECULAR PROPERTY REGRESSION

## ABSTRACT

Molecular property regression often suffers from severe target distribution imbalance: models tend to overfit to dense regions while underperforming on rare but critical ones. This challenge arises from the continuous-valued nature and complex structure–property relationships of molecular datasets, where molecules with highly dissimilar structures may exhibit similar properties. These characteristics pose challenges to many existing imbalance-handling methods, limiting their effectiveness when applied to molecular regression tasks. We propose **Distribution-Guided Expert Routing (DistRouting)**, a flexible framework that dynamically assigns molecules to specialized experts based on predicted target ranges. Routing decisions integrate deep molecular embeddings and physicochemical descriptors to better reflect both learned representations and domain knowledge. To enhance robustness, DistRouting employs a soft Top-$k$ routing strategy, enabling each sample to attend to multiple experts. We incorporate DistRouting as a plug-in module into four representative models and evaluate it on multiple molecular property prediction benchmarks. Our approach consistently improves performance in rare target regions, demonstrating its effectiveness in addressing label imbalance in molecular regression tasks.

## 1 INTRODUCTION

Accurately predicting molecular properties is a fundamental problem in computational chemistry and drug discovery (Gilmer et al., 2017; Wu et al., 2018; Yang et al., 2019). Many molecular property prediction tasks are formulated as regression problems, where the goal is to estimate continuous-valued outcomes such as binding affinity, solubility, and toxicity. However, these tasks often suffer from imbalanced target distributions, where most data points concentrate in a narrow target range while chemically significant outliers are sparsely distributed. This distributional imbalance causes standard regression models to focus on dense target regions while neglecting rare but critical ones.

Prevailing approaches for handling imbalanced regression can be broadly categorized into data resampling, loss reweighting, and feature-level calibration. SMOGN (Branco et al., 2017) represents a resampling strategy that interpolates low-density regions, yet such methods are generally unsuitable for structured molecular data, as naive interpolation in high-dimensional molecular graphs often yields invalid or unrealistic samples that violate chemical constraints (You et al., 2020; Rong et al., 2020). Loss reweighting methods such as DenseWeight (Steininger et al., 2021) and label distribution smoothing (LDS) (Yang et al., 2021), as well as feature distribution smoothing (FDS) (Yang et al., 2021), adjust sample importance or smooth representations based on target density. While effective in general-purpose regression, these methods share a common assumption that samples with similar labels are also close in the input space. However, molecular datasets often violate this assump-

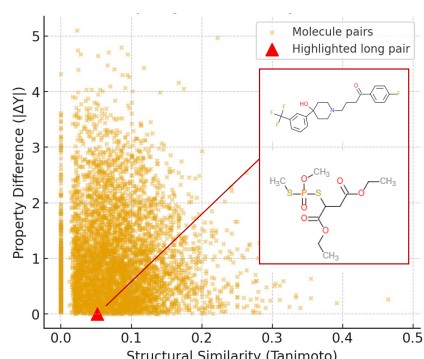

Figure 1: Structure–property mismatch in the LD50 dataset. Molecules with highly dissimilar structures can share identical property.

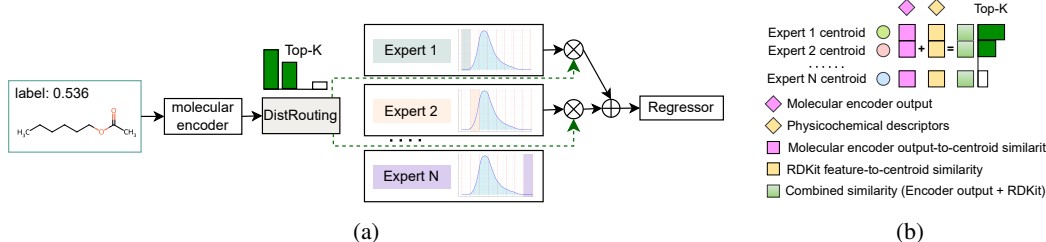

Figure 2: **DistRouting framework and routing mechanism.** (a) **Overall architecture.** DistRouting selects Top-$k$ experts by similarity and aggregates outputs for molecular property prediction. (b) **Routing mechanism.** Top-$k$ selection combines similarity scores between expert centroids and both encoder outputs and RDKit physicochemical descriptors.

tion due to complex structure–property relationships: as shown in Figure 1, structurally dissimilar molecules may exhibit identical properties. In such cases, reweighting methods may overemphasize rare samples, improving the fit to those molecules but failing to generalize to structurally diverse compounds with similar labels. In contrast, feature calibration approaches smooth embeddings based on label similarity, but risk blurring critical distinctions by forcing structurally distinct yet label-similar molecules closer in the representation space. These limitations undermine the ability of the model to capture true structure–property relationships and ultimately reduce generalization.

To address these issues, we approach the challenge from the perspective of model architecture and propose **DistRouting**, a distribution-guided expert routing framework for imbalanced molecular property regression. DistRouting partitions the target space into intervals and assigns specialized experts to each, enabling the model to capture interval-level commonalities while preserving structural diversity. Rather than relying on shared parameters that bias optimization toward high-frequency labels, DistRouting allocates experts to different regions, ensuring that rare but mechanistically distinct compounds are not overwhelmed by dominant ones. As illustrated in Figure 2a, routing decisions are guided jointly by molecular embeddings and RDKit-based physicochemical descriptors (Landrum, 2006), while a soft Top-$k$ strategy allows each molecule to attend to multiple experts, improving robustness in sparse regions. Furthermore, we align routing behavior with the target distribution via a KL divergence loss on soft interval labels and introduce an interval-aware contrastive loss to structure the embedding space. By explicitly linking label distribution to representation learning, DistRouting enables rare samples to share parameters within appropriate experts, alleviating their isolation and yielding more effective modeling of imbalanced molecular property data.

Our main contributions are summarized as follows:

- We propose **DistRouting**, a distribution-guided expert routing framework for molecular property regression under imbalanced target distributions. By routing samples to specialized experts assigned to different target ranges, DistRouting effectively addresses the often-overlooked challenge of target imbalance and consistently improves the performance of diverse molecular encoders, with particularly strong gains in rare target regions.

- We introduce a **physicochemical descriptors–guided routing mechanism**, which leverages RDKit-derived physicochemical descriptors to assist in expert selection. These features act as chemically informed priors that stabilize routing decisions.

- We further propose an **interval-aware supervised contrastive learning loss** to structure the molecular representation space, promoting semantic alignment among samples within the same target interval to facilitate consistent expert routing.

## 2 RELATED WORK

**Imbalance Regression.** Existing approaches can be broadly categorized into three groups: data resampling, loss reweighting, and feature-level calibration. Most existing work adapts the SMOTE algorithm to regression (Blagus & Lusa, 2013; Branco et al., 2017; 2018), where synthetic samples are generated for rare target regions by interpolation or by adding noise. Loss reweighting strategies,

including DenseWeight (Steininger et al., 2021) and label distribution smoothing (LDS) (Yang et al., 2021), adjust the training objective by assigning larger weights to samples from underrepresented target regions, effectively biasing optimization toward rare values. Feature distribution smoothing (FDS) (Yang et al., 2021) instead calibrates hidden representations by smoothing features according to target density, mitigating overfitting to noisy or sparse regions.

**Molecular Encoders.** Learning effective molecular representations is fundamental to property prediction. Graph Neural Networks (GNNs) model molecules as atom–bond graphs (Gilmer et al., 2017; Xu et al., 2018; Hu et al., 2020), with variants such as Graph Attention Networks (GAT) (Velickovic et al., 2017) and DeeperGCN (Li et al., 2023) improving message passing via attention and residual connections. In parallel, sequence-based encoders process SMILES strings as molecular language, with transformer models such as ChemBERTa (Chithrananda et al., 2020), SMILES-BERT (Wang et al., 2019), and Chemformer (Irwin et al., 2022) demonstrating strong performance through large-scale self-supervised pretraining. Extending to 3D molecular structures, UniMol (Zhou et al., 2023) provides a unified framework that captures richer spatial information. In our work, we evaluate **DistRouting** as a plug-in module across these modalities, showing consistent gains for diverse molecular encoders.

**Mixture of Experts.** Mixture-of-Experts (MoE) architectures (Jacobs et al., 1991; Shazeer et al., 2017; Dai et al., 2024) route each input to a sparse subset of experts via a gating mechanism. Each expert is an independent feed-forward network, and the selected outputs of experts are combined with learned weights. In our work, we adopt this framework but make experts distribution-aware, so that each specializes in a target interval of the regression space.

**Supervised Contrastive Learning.** Contrastive learning aims to learn representations by pulling similar samples closer and pushing dissimilar ones apart (Chen et al., 2020). Supervised contrastive learning (SCL) (Khosla et al., 2020) extends this paradigm by leveraging label information, so that samples with the same label are treated as positives. We further adapt this idea to **Interval-Aware SCL (ISCL)**, where positives are defined as samples within the same target interval and negatives otherwise. This encourages the learned embeddings to align with expert assignments.

## 3 METHOD: DISTROUTING

### 3.1 PRELIMINARIES AND NOTATION

Let $\{(x_i, y_i)\}_{i=1}^N$ denote the training set, where $x_i \in \mathbb{R}^d$ is the input molecule and $y_i \in \mathbb{R}$ is the corresponding continuous-valued molecular property. We partition the label space $\mathcal{Y}$ into $B$ intervals $\{I_1, \ldots, I_B\}$, each defined by boundaries $[y_{b-1}, y_b]$, where $b \in \mathcal{B} = \{1, \ldots, B\}$ indexes the intervals. Each interval $I_b$ is associated with an interval center $c_b$, which is used for routing supervision and representation guidance. We assign a dedicated expert to each interval, enabling specialization across different target ranges. Given an input molecule $x$, we use a molecular encoder $f(x; \theta)$ to extract a representation $z \in \mathbb{R}^d$. Additionally, we extract 200 physicochemical descriptors from RDKit and map them to $\mathbb{R}^d$ via a multilayer perceptron (MLP) to obtain a descriptor embedding $r \in \mathbb{R}^d$, which is used to guide the expert routing process.

### 3.2 DISTRIBUTION-AWARE ROUTING GUIDED BY PHYSICOCHEMICAL DESCRIPTORS

As illustrated in Figure 2b, we design a hybrid routing mechanism that leverages molecular embeddings and physicochemical descriptors to assign molecules to experts in a distribution-aware manner. Given a molecular embedding $z$ and an RDKit-derived descriptor vector $r$, each expert, indexed by its interval $b \in \{1, \ldots, B\}$, is associated with a learnable centroid vector $e_b \in \mathbb{R}^d$.

We compute the combined similarity between the input features and each expert centroid as:

$$s_b = z^\top e_b + r^\top e_b. \tag{1}$$

Routing weights are then computed by applying a softmax over the score vector $\{s_b\}$ and retaining only the top-$k$ experts:

$$
g_b = \begin{cases} \text{softmax}_b(s), & \text{if } b \in \text{TopK}(s, K), \\ 0, & \text{otherwise,} \end{cases} \tag{2}
$$

where $\text{softmax}_b(s)$ denotes the $b$-th component of the softmax applied over all scores $\{s_1, \ldots, s_B\}$.

The final output is obtained by aggregating the responses from the selected experts:

$$
z' = \sum_{b=1}^{B} g_b \cdot \text{FFN}_b(z), \tag{3}
$$

where $\text{FFN}_b(\cdot)$ denotes the $b$-th expert network.

### 3.3 GATING SUPERVISION WITH SOFT TARGET LABELS

As described in the problem setting, the continuous target space $\mathcal{Y}$ is partitioned into $B$ intervals $\{I_1, I_2, \ldots, I_B\}$, each with a centroid $c_b$ and width $w_b$. This structure enables us to encode coarse-grained semantics over target values, which we leverage to supervise expert routing.

Given a sample $(x_i, y_i)$, we compute a soft target vector $\mathbf{q}_i \in \mathbb{R}^B$ indicating the degree to which the target $y_i$ belongs to each interval. The assignment is based on a normalized Gaussian kernel, scaled by the width of each interval:

$$
\mathbf{q}_{ib} = \frac{\exp\left(-\frac{1}{2}\left(\frac{y_i - c_b}{w_b \cdot \sigma}\right)^2\right)}{\sum\limits_{j=1}^{B} \exp\left(-\frac{1}{2}\left(\frac{y_i - c_j}{w_j \cdot \sigma}\right)^2\right)}, \tag{4}
$$

where $\sigma$ is a temperature hyperparameter controlling the smoothness of the soft labels. The predicted routing distribution $\mathbf{p}_i \in \mathbb{R}^B$ is computed by applying softmax over the expert scores $\mathbf{s}_i = \{s_{i1}, \ldots, s_{iB}\}$, where $s_{ib}$ is the similarity score between sample $i$ and expert $b$. The gating loss is then defined as:

$$
\mathcal{L}_{\text{gate}} = \text{KL}(\mathbf{q}_i \,\|\, \mathbf{p}_i). \tag{5}
$$

### 3.4 INTERVAL-AWARE SUPERVISED CONTRASTIVE LEARNING

To promote semantic structure and expert specialization, we propose an interval-aware supervised contrastive learning (ISCL) loss. Unlike standard contrastive learning with discrete labels, ISCL handles continuous regression targets by grouping molecules into intervals and assigning soft labels based on target proximity.

While routing assigns molecules to experts by feature similarity, ISCL complements this by pulling together samples with similar soft labels and pushing apart dissimilar ones, applied jointly to molecular embeddings and physicochemical descriptors (Figure 3).

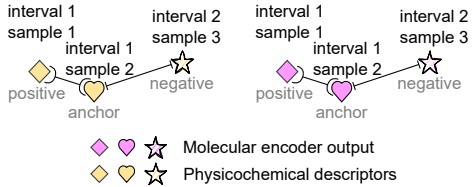

Figure 3: **ISCL mechanism.** ISCL constructs positives within the same interval and negatives across intervals, applied to both molecular embeddings and physicochemical descriptors.

Specifically, for each sample $x_i$ with target $y_i$, we compute a soft target vector $\mathbf{q}_i \in \mathbb{R}^B$ using a normalized Gaussian kernel as described in Section A.4.5. Given a minibatch of samples $\{(x_i, y_i)\}_{i=1}^{M}$, we extract two representations per sample: the molecular encoder output $z_i = f(x_i; \theta)$ and the corresponding RDKit-derived descriptor embedding $r_i \in \mathbb{R}^d$. ISCL is applied independently to both views, and the total contrastive loss is:

$$
\mathcal{L}_{\text{ISCL}} = \mathcal{L}_{\text{ISCL}}^{\text{mol}} + \mathcal{L}_{\text{ISCL}}^{\text{rdkit}}, \tag{6}
$$

Each component follows a weighted supervised contrastive formulation:

$$\mathcal{L}_{\text{ISCL}}^{(\cdot)} = -\frac{1}{|I|} \sum_{i \in I} \frac{1}{|P(i)|} \sum_{j \in P(i)} \log \frac{\exp\left(\text{sim}(v_i, v_j)/\tau\right)}{\sum_{k \neq i} \exp\left(\text{sim}(v_i, v_k)/\tau\right) \cdot w_{ik}}, \tag{7}$$

where $v_i$ is either $z_i$ or $r_i$, and the weighting term is defined as:

$$w_{ik} = \gamma^{1-\cos(\mathbf{q}_i, \mathbf{q}_k)}, \tag{8}$$

with $\gamma > 1$ controlling the penalty strength and where $\cos(\cdot)$ denotes cosine similarity. This encourages stronger repulsion between samples from dissimilar target intervals.

## 3.5 Prediction and Overall Loss

Following expert routing and aggregation, the fused representation is passed through an MLP head to obtain the final prediction. The overall training objective combines three components:

$$\mathcal{L} = \mathcal{L}_{\text{reg}} + \mathcal{L}_{\text{gate}} + \lambda \cdot \mathcal{L}_{\text{ISCL}}, \tag{9}$$

where $\mathcal{L}_{\text{reg}}$ is the mean squared error loss, $\mathcal{L}_{\text{gate}}$ is the KL divergence loss supervising expert routing (Section A.4.5), and $\mathcal{L}_{\text{ISCL}}$ is the interval-aware supervised contrastive loss (Section 3.4). The coefficient $\lambda$ balances the contrastive objective. A full training procedure is provided in Appendix A.1.

## 4 Experiments

### 4.1 Experimental Setting

**Datasets.** We evaluate DistRouting on five molecular property regression benchmarks from the Therapeutics Data Commons (TDC) Huang et al. (2021): Caco2_Wang, Lipophilicity_AstraZeneca, PPBR_AZ, LD50_Zhu, and QM9. These datasets cover diverse biophysical and pharmacokinetic properties and exhibit target imbalance (details in Appendix A.2). For the first four datasets, we adopt standard 5-fold scaffold split of TDC, with each fold divided into 70% training, 10% validation, and 20% test. For QM9, we use the provided random split and evaluate the HOMO–LUMO gap.

**Backbone Models.** We evaluate DistRouting as a generic plug-in module across multiple standard encoders. We consider five representative backbone models: GAT (Velickovic et al., 2017), DeeperGCN (Li et al., 2023), ChemBERTa (Chithrananda et al., 2020), GROVER (Rong et al., 2020), and UniMol (Zhou et al., 2023), where UniMol is used only for QM9 as it leverages 3D molecular structures for prediction. Each backbone is compared with its corresponding DistRouting-enhanced variant. Implementation details of each encoder are provided in Appendix A.3.

**Evaluation Process and Details.** We evaluate model performance using both mean absolute error (MAE) and Pearson correlation coefficient (PCC). To further examine robustness under target imbalance, we report region-wise MAE and PCC on the head and tail intervals, defined as the bottom and top 20% quantiles of the target distribution. All results are reported with mean and standard deviation over 5 splits. The target space is partitioned into $B = 8$ intervals, and for each sample, the Top-$k = 2$ most relevant experts are selected via similarity-guided routing. The soft label smoothing parameter $\sigma$ in Eq. 4 is set to 0.7. All models are trained using the Adam optimizer with a learning rate of $1e-4$ and a batch size of 128, with the best model selected based on validation MAE. Additional hyperparameter settings are provided in Appendix A.3.5.

### 4.2 Overall Performance

Figure 10 (Appendix A.4.1) shows the validation MAE curves across training epochs, where DistRouting consistently achieves lower errors. Tables 1 and 9 (Appendix A.4.2) present the test results in terms of MAE and PCC, comparing each vanilla backbone with its DistRouting-enhanced counterpart. Across all backbones, incorporating DistRouting leads to consistently better overall performance, with the best results on Lipophilicity and LD50 achieved by GROVER + DistRouting, including a substantial reduction on LD50 from 0.684 to 0.539.

Table 1: MAE (↓) on the four datasets. Bold numbers indicate the best result in each column.

| Method | Caco2 | Lipophilicity | PPBR | LD50 |
|---|---|---|---|---|
| DeeperGCN | $0.366 \pm 0.012$ | $0.528 \pm 0.012$ | $8.355 \pm 0.211$ | $0.643 \pm 0.010$ |
| DeeperGCN + DistRouting | $\mathbf{0.315 \pm 0.009}$ | $\mathbf{0.509 \pm 0.013}$ | $\mathbf{7.849 \pm 0.028}$ | $\mathbf{0.616 \pm 0.009}$ |
| GAT | $0.383 \pm 0.012$ | $0.607 \pm 0.008$ | $7.940 \pm 0.148$ | $0.651 \pm 0.014$ |
| GAT + DistRouting | $\mathbf{0.327 \pm 0.011}$ | $\mathbf{0.551 \pm 0.010}$ | $\mathbf{7.780 \pm 0.154}$ | $\mathbf{0.609 \pm 0.026}$ |
| ChemBERTa | $0.352 \pm 0.019$ | $0.568 \pm 0.009$ | $8.069 \pm 0.122$ | $0.651 \pm 0.009$ |
| ChemBERTa + DistRouting | $\mathbf{0.329 \pm 0.011}$ | $\mathbf{0.548 \pm 0.007}$ | $\mathbf{7.869 \pm 0.131}$ | $\mathbf{0.614 \pm 0.014}$ |
| GROVER | $0.393 \pm 0.008$ | $0.516 \pm 0.031$ | $9.180 \pm 0.340$ | $0.684 \pm 0.046$ |
| GROVER + DistRouting | $\mathbf{0.358 \pm 0.013}$ | $\mathbf{0.450 \pm 0.009}$ | $\mathbf{8.188 \pm 0.378}$ | $\mathbf{0.539 \pm 0.025}$ |

## 4.3 REGION-WISE PERFORMANCE

Table 2: Region-wise MAE (↓) across all datasets and methods. Best between each pair is bolded.

| Dataset | Method | Head MAE ↓ | Body MAE ↓ | Tail MAE ↓ |
|---|---|---|---|---|
| Caco2 | DeeperGCN | $0.359 \pm 0.031$ | $0.376 \pm 0.015$ | $0.354 \pm 0.041$ |
| | DeeperGCN + DistRouting | $\mathbf{0.306 \pm 0.025}$ | $\mathbf{0.341 \pm 0.032}$ | $\mathbf{0.267 \pm 0.039}$ |
| | GAT | $0.524 \pm 0.031$ | $\mathbf{0.298 \pm 0.009}$ | $0.350 \pm 0.054$ |
| | GAT + DistRouting | $\mathbf{0.285 \pm 0.039}$ | $0.358 \pm 0.041$ | $\mathbf{0.323 \pm 0.030}$ |
| | ChemBERTa | $0.403 \pm 0.025$ | $\mathbf{0.332 \pm 0.032}$ | $0.313 \pm 0.021$ |
| | ChemBERTa + DistRouting | $\mathbf{0.359 \pm 0.026}$ | $0.344 \pm 0.007$ | $\mathbf{0.238 \pm 0.029}$ |
| | GROVER | $0.476 \pm 0.076$ | $\mathbf{0.333 \pm 0.044}$ | $0.403 \pm 0.039$ |
| | GROVER + DistRouting | $\mathbf{0.366 \pm 0.023}$ | $0.375 \pm 0.009$ | $\mathbf{0.304 \pm 0.077}$ |
| Lipophilicity | DeeperGCN | $0.656 \pm 0.029$ | $0.449 \pm 0.012$ | $0.638 \pm 0.037$ |
| | DeeperGCN + DistRouting | $\mathbf{0.597 \pm 0.022}$ | $\mathbf{0.445 \pm 0.015}$ | $\mathbf{0.614 \pm 0.022}$ |
| | GAT | $0.881 \pm 0.035$ | $\mathbf{0.457 \pm 0.013}$ | $0.782 \pm 0.047$ |
| | GAT + DistRouting | $\mathbf{0.673 \pm 0.032}$ | $0.464 \pm 0.006$ | $\mathbf{0.691 \pm 0.064}$ |
| | ChemBERTa | $0.719 \pm 0.017$ | $0.528 \pm 0.015$ | $\mathbf{0.539 \pm 0.016}$ |
| | ChemBERTa + DistRouting | $\mathbf{0.711 \pm 0.017}$ | $\mathbf{0.463 \pm 0.017}$ | $0.639 \pm 0.036$ |
| | GROVER | $0.612 \pm 0.088$ | $0.402 \pm 0.023$ | $0.758 \pm 0.078$ |
| | GROVER + DistRouting | $\mathbf{0.512 \pm 0.038}$ | $\mathbf{0.393 \pm 0.013}$ | $\mathbf{0.557 \pm 0.043}$ |
| PPBR | DeeperGCN | $19.671 \pm 0.398$ | $6.041 \pm 0.282$ | $3.923 \pm 0.436$ |
| | DeeperGCN + DistRouting | $\mathbf{19.285 \pm 0.425}$ | $\mathbf{5.633 \pm 0.095}$ | $\mathbf{2.992 \pm 0.232}$ |
| | GAT | $\mathbf{18.541 \pm 0.997}$ | $\mathbf{5.591 \pm 0.431}$ | $4.346 \pm 0.341$ |
| | GAT + DistRouting | $19.142 \pm 1.801$ | $5.722 \pm 0.575$ | $\mathbf{2.515 \pm 0.368}$ |
| | ChemBERTa | $20.810 \pm 0.861$ | $\mathbf{5.580 \pm 0.274}$ | $2.723 \pm 0.425$ |
| | ChemBERTa + DistRouting | $\mathbf{18.364 \pm 0.913}$ | $6.084 \pm 0.307$ | $\mathbf{2.654 \pm 0.601}$ |
| | GROVER | $18.666 \pm 0.768$ | $6.934 \pm 0.620$ | $6.401 \pm 0.565$ |
| | GROVER + DistRouting | $\mathbf{17.987 \pm 1.456}$ | $\mathbf{6.337 \pm 0.708}$ | $\mathbf{3.882 \pm 1.264}$ |
| LD50 | DeeperGCN | $0.513 \pm 0.033$ | $\mathbf{0.454 \pm 0.022}$ | $1.338 \pm 0.053$ |
| | DeeperGCN + DistRouting | $\mathbf{0.500 \pm 0.019}$ | $0.465 \pm 0.024$ | $\mathbf{1.182 \pm 0.062}$ |
| | GAT | $0.495 \pm 0.028$ | $0.445 \pm 0.004$ | $1.423 \pm 0.074$ |
| | GAT + DistRouting | $\mathbf{0.471 \pm 0.020}$ | $\mathbf{0.438 \pm 0.017}$ | $\mathbf{1.258 \pm 0.153}$ |
| | ChemBERTa | $0.458 \pm 0.048$ | $0.449 \pm 0.010$ | $1.447 \pm 0.052$ |
| | ChemBERTa + DistRouting | $\mathbf{0.441 \pm 0.029}$ | $\mathbf{0.431 \pm 0.013}$ | $\mathbf{1.315 \pm 0.069}$ |
| | GROVER | $0.673 \pm 0.093$ | $0.405 \pm 0.017$ | $1.530 \pm 0.201$ |
| | GROVER + DistRouting | $\mathbf{0.502 \pm 0.045}$ | $\mathbf{0.397 \pm 0.007}$ | $\mathbf{1.005 \pm 0.102}$ |

Table 2 and Figure 11 (Appendix A.4.3) present MAE performance in the **head**, **body**, and **tail** regions across all datasets, comparing each backbone model with its DistRouting-enhanced version. Across the four datasets and four backbones, the **head** and **tail** regions comprise a total of 32 evaluations, among which DistRouting achieves improvements in 30 cases. In several settings the gains are particularly pronounced, such as GAT on the head region of Caco2 ($0.524 \rightarrow 0.285$) and GROVER on the tail region of Lipophilicity ($0.758 \rightarrow 0.557$).

The body region remains stable across models, with DistRouting showing modest improvements or parity. This suggests that substantial gains in the head and tail regions are achieved without compromising performance in the dense body. Overall, the results highlight DistRouting's ability to improve generalization, especially in underrepresented regions where baseline models struggle.

## 4.4 ABLATION STUDY

Table 3: Ablation study of DistRouting components. Each row corresponds to a variant with specific modules removed. ✓ indicates the component is used. We report MAE (↓) on four datasets.

| Variant | MoE Routing | RDKit | Gate Sup. | ISCL | Caco2 ↓ | PPBR ↓ | Lipo ↓ | LD50 ↓ |
|---|---|---|---|---|---|---|---|---|
| Full Model | ✓ | ✓ | ✓ | ✓ | **0.315** | **7.849** | 0.509 | 0.616 |
| w/o Gating Sup. | ✓ | ✓ | ✗ | ✓ | 0.343 | 8.145 | 0.510 | 0.632 |
| w/o ISCL | ✓ | ✓ | ✓ | ✗ | 0.322 | 7.859 | 0.523 | **0.608** |
| w/o RDKit Guidance | ✓ | ✗ | ✓ | ✓ | 0.340 | 8.266 | **0.507** | 0.637 |
| MoE Routing only | ✓ | ✗ | ✗ | ✗ | 0.343 | 8.314 | 0.530 | 0.641 |

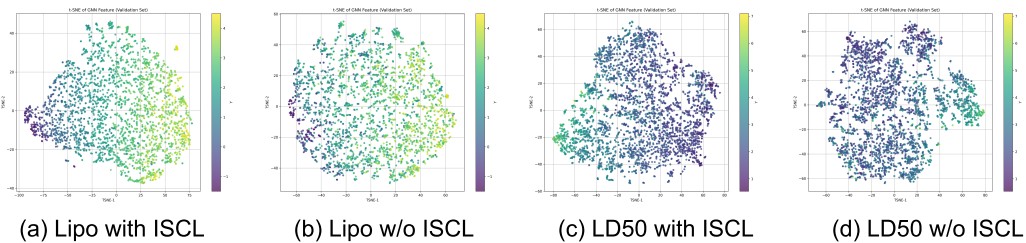

(a) Lipo with ISCL     (b) Lipo w/o ISCL     (c) LD50 with ISCL     (d) LD50 w/o ISCL

Figure 4: **t-SNE visualization of molecular representations.** Molecular embeddings for the Lipophilicity and LD50 datasets under two settings: with ISCL (a, c) and without ISCL (b, d). Representations are extracted after the molecular encoder and before expert routing.

Based on the ablation results in Table 3, conducted with DeeperGCN as the backbone, we find that gating supervision is the most critical component: its removal consistently degrades performance across all datasets. The impact of ISCL, in contrast, is less consistent. While it brings clear improvements on the Lipophilicity dataset, its effect is minimal on others and even slightly detrimental on LD50. Nevertheless, t-SNE visualizations in Figure 4 suggest that ISCL meaningfully enhances the structure of the representation space. This observation is further corroborated by the quantitative alignment metrics reported in Table 10 (Appendix A.4.4). We further examined ISCL by varying its loss weight (Table 11, Appendix A.4.6). Performance degrades with small weights on LD50 but recovers as the weight increases, reaching an MAE of 0.605 at $\lambda = 3$. This suggests that weak ISCL signals are insufficient and may conflict with the gating objective. Ablations on auxiliary supervision show similar trends: removing RDKit guidance causes moderate drops but still outperforms the baseline encoder, indicating that the routing mechanism provides useful inductive bias. In contrast, the MoE-only variant performs the worst, highlighting that expert specialization requires distribution-aware guidance.

## 5 DISCUSSION

Table 4: Comparison of imbalance handling methods on the DeeperGCN backbone. DistRouting achieves the lowest MAE across datasets.

| Method | Caco2 | Lipophilicity | PPBR | LD50 |
|---|---|---|---|---|
| DeeperGCN | $0.366 \pm 0.012$ | $0.528 \pm 0.012$ | $8.355 \pm 0.211$ | $0.643 \pm 0.010$ |
| + DenseWeight | $0.383 \pm 0.032$ | $0.577 \pm 0.011$ | $10.418 \pm 0.859$ | $0.691 \pm 0.033$ |
| + FDS | $0.365 \pm 0.016$ | $0.605 \pm 0.006$ | $8.955 \pm 0.249$ | $0.700 \pm 0.020$ |
| + LDS | $0.409 \pm 0.053$ | $0.553 \pm 0.014$ | $10.161 \pm 0.303$ | $0.691 \pm 0.024$ |
| + DistRouting (ours) | $\mathbf{0.315 \pm 0.009}$ | $\mathbf{0.509 \pm 0.013}$ | $\mathbf{7.849 \pm 0.028}$ | $\mathbf{0.616 \pm 0.009}$ |

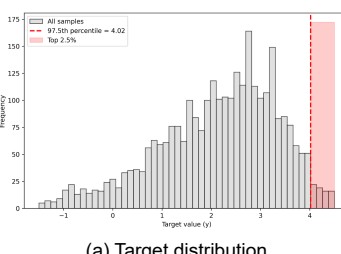
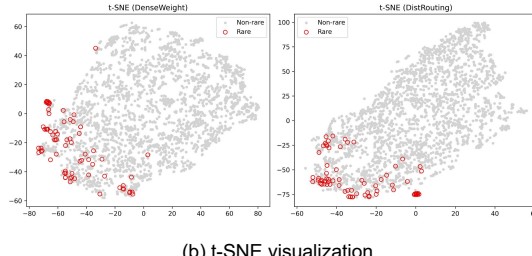

(a) Target distribution     (b) t-SNE visualization

Figure 5: **Rare-sample analysis on the Lipophilicity dataset.** (a) Target distribution with the top 2.5% highlighted. (b) t-SNE visualization of rare-sample embeddings. Under DenseWeight (left), rare samples show partial clustering but many remain as outliers in the embedding space. In contrast, DistRouting (right) yields more compact and coherent clusters.

**Comparison with Existing Imbalance Regression Methods.** We further compare DistRouting with representative imbalance regression approaches, including DenseWeight (Steininger et al., 2021), FDS and LDS (Yang et al., 2021), all implemented on the DeeperGCN backbone. As shown in Table 4, these methods generally degrade performance relative to vanilla DeeperGCN.

We visualize the learned embeddings of rare samples (with target values in the top 2.5%) using t-SNE. Figure 5 compares DenseWeight and DistRouting. DistRouting forms more compact clusters, providing empirical support for our hypothesis that while reweighting increases the weights of structurally diverse molecules with similar labels, it does not ensure that these samples share representations in the model and they may remain isolated, thereby limiting generalization.

**Parameter analysis.** To examine whether performance gains stem from parameter counts, we compare the baseline 2-layer MLP with larger ones by increasing hidden size or depth on the LD50 dataset. As shown in Table 5, simply enlarging the MLP fails to improve MAE and even degrades performance, indicating that DistRouting's improvements arise from its routing mechanism rather than model scale.

Table 5: MAE performance of baseline and larger MLP models on the LD50 dataset.

| Setting | MAE |
|---|---|
| Baseline (2-layer MLP, hidden=512) | $0.643 \pm 0.010$ |
| 2-layer MLP (hidden=2048) | $0.649 \pm 0.017$ |
| 2-layer MLP (hidden=4096) | $0.808 \pm 0.003$ |
| 4-layer MLP (hidden=512) | $0.803 \pm 0.002$ |

**Distribution Matching.** To quantitatively assess how well each model captures the overall target distribution, we compute the Jensen–Shannon (JS) distance between the predicted and true value distributions on the test sets (see Appendix A.5 for details). As shown in Table 6, when implemented on the DeeperGCN backbone, DistRouting achieves lower

Table 6: JS distance ($\downarrow$) between predicted and true target distributions.

| Method | Caco2 | PPBR | LD50 | Lipo |
|---|---|---|---|---|
| DeeperGCN | 0.1530 | 0.1417 | 0.1872 | 0.1508 |
| + DistRouting | **0.1265** | **0.1233** | **0.1444** | **0.1342** |

JS distances than the baseline across all four datasets, indicating better global alignment with the true label distribution. Figure 6 compares the baseline encoder with its DistRouting-enhanced version. The baseline shows a central bias, while DistRouting better captures head and tail regions, consistent with the lower MAE reported in Table 2.

**Gating Behavior Analysis.** To assess the effect of gating supervision, Figure 7 shows expert assignments across target values on the LD50 dataset. With KL-based supervision (Figure 7a), experts specialize in distinct regions of the target space, forming a structured partition aligned with interval semantics. Without supervision (Figure 7b), routing becomes disorganized: experts collapse to overlapping or narrow ranges, and some remain unused.

This comparison reveals that the gating supervision plays a critical role in promoting expert diversity and enforcing consistent expert-target alignment. Without this loss, the model struggles to utilize expert capacity effectively. These findings align with the ablation results in Table 3, where removing

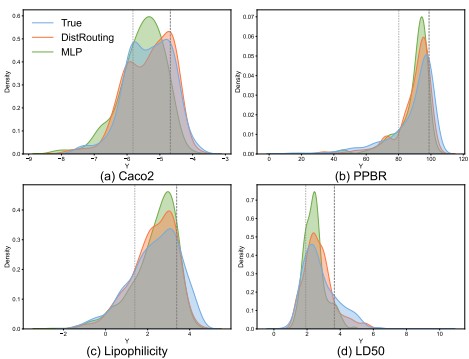
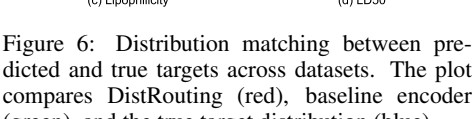
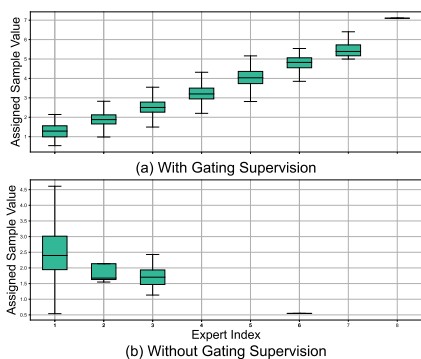

Figure 6: Distribution matching between predicted and true targets across datasets. The plot compares DistRouting (red), baseline encoder (green), and the true target distribution (blue).

Figure 7: Distribution of expert assignments across target values on the LD50 dataset: (a) with gating supervision; (b) without gating supervision.

gating supervision leads to a significant performance drop. Similar trends are observed on other datasets, as shown in Figure 12 (Appendix A.4.5).

**Effect of Number of Experts.** We extended the ablation study on the Lipophilicity dataset with 2, 4, 6, 8, 10, 12, 16, 20, and 30 experts. Comparable performance is observed for 6–16 experts, while using only 2 experts or increasing to 20–30 experts leads to a noticeable MAE increase (Figure 8).

Performance degradation with 2 experts arises from insufficient specialization: with Top-$k = 2$, each input aggregates outputs from both experts, weakening selective routing and diminishing the benefit of targeted specialization. Excessive experts also harm performance, likely due to fragmentation and underutilization. These results confirm that the effectiveness of DistRouting stems from targeted specialization rather than increased model capacity.

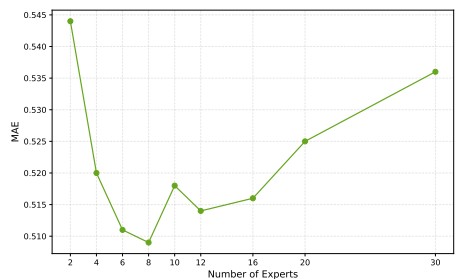

Figure 8: Effect of number of experts on MAE for the Lipophilicity dataset.

**Generalization to Large Datasets.** To assess scalability and generalization, we evaluated DistRouting on QM9, a large-scale molecular benchmark. As shown in Table 12 (Appendix A.4.7), incorporating DistRouting into UniMol reduces MAE from 0.0084 to 0.0066 and raises PCC from 0.9690 to 0.9790, demonstrating clear gains. Region-wise analysis of the HOMO–LUMO gap (Table 13) further shows consistent MAE reductions across head, body, and tail regions. These results indicate that DistRouting captures both frequent and rare targets, underscoring its robustness and ability to generalize beyond smaller datasets.

## 6 CONCLUSION

We presented **DistRouting**, a distribution-guided expert routing framework that addresses molecular property regression with imbalanced targets through architectural specialization. By assigning samples to experts for different target ranges and incorporating RDKit-guided routing with an interval-aware contrastive loss, DistRouting improves performance across diverse encoders, especially in rare regions.

**Limitations.** DistRouting currently relies on uniform interval partitioning of the target space. Future work could consider property-aware partitioning strategies that incorporate the semantic meaning of molecular properties to better guide expert specialization.

ETHICS STATEMENT

This work develops **DistRouting**, a distribution-aware expert routing framework for molecular property regression under imbalanced targets. Its applications mainly lie in computational chemistry and drug discovery, which are intended to advance scientific understanding and provide societal benefits. We do not foresee immediate negative ethical risks. We encourage the responsible use of artificial intelligence in biomedical and chemical research.

REPRODUCIBILITY STATEMENT

We have provided detailed descriptions of model architecture, training procedures, and datasets. The source code, configuration files, and scripts necessary to reproduce our results will be released in a public repository upon publication.

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

## A   APPENDIX

### A.1   PSEUDOCODE FOR DISTROUTING

The following algorithm summarizes the training procedure of DistRouting, which combines distribution-aware routing, physicochemical priors, and contrastive supervision.

---

**Algorithm 1:** DistRouting: Distribution-Aware Expert Routing with Physicochemical Guidance

---

**Input:** Training data $\mathcal{D} = \{(x_i, y_i)\}_{i=1}^N$;
     Molecular encoder $f(\cdot; \theta)$; RDKit extractor $\text{rdk}(\cdot)$;
     RDKit encoder $\text{enc}_{\text{rdk}}(\cdot)$;
     Number of experts $B$; top-$k$ routing $k$;
     Temperature parameters $\tau, \sigma$; ISCL weight $\lambda$
**Output:** Trained encoder $f(\cdot)$ and expert networks $\{\text{FFN}_j\}_{j=1}^B$

1   Partition target space $\mathcal{Y}$ into $B$ intervals $\{I_1, \ldots, I_B\}$ with centers $\{c_1, \ldots, c_B\}$ and widths $\{w_1, \ldots, w_B\}$;

2   Initialize expert centroids $\{e_j\}_{j=1}^B$;

3   **while** *not converged* **do**

4      Sample minibatch $\{(x_i, y_i)\}_{i=1}^M$;

5      **foreach** $x_i$ *in minibatch* **do**

6         Compute molecular embedding: $z_i \leftarrow f(x_i)$;

7         Compute RDKit descriptors and encode: $r_i \leftarrow \text{enc}_{\text{rdk}}(\text{rdk}(x_i))$;

8         **foreach** *expert* $j = 1$ *to* $B$ **do**

9            Compute similarity: $s_{ij} \leftarrow \text{sim}(z_i, e_j) + \text{sim}(r_i, e_j)$;

10        Compute routing weights: $\alpha_{ij} \leftarrow \text{softmax}_j(s_{ij})$;

11        Construct sparse gate: $g_{ij} \leftarrow \alpha_{ij}$ if $j \in \text{TopK}(\alpha_i, k)$ else 0;

12        Expert output: $h_i \leftarrow \sum_{j=1}^B g_{ij} \cdot \text{FFN}_j(z_i)$;

13        Final prediction: $\hat{y}_i \leftarrow \text{MLP}(h_i)$;

14        Compute soft routing label:

$$w_{ij} \leftarrow \frac{\exp\left(-\frac{1}{2}\left(\frac{y_i - c_j}{w_j \cdot \sigma}\right)^2\right)}{\sum_{l=1}^B \exp\left(-\frac{1}{2}\left(\frac{y_i - c_l}{w_l \cdot \sigma}\right)^2\right)}$$

15      Compute losses:

16        Regression loss: $\mathcal{L}_{\text{reg}} \leftarrow \text{MSE}(\hat{y}_i, y_i)$;

17        Gating loss: $\mathcal{L}_{\text{gate}} \leftarrow \text{KL}(w_i \,\|\, \alpha_i)$;

18        Contrastive loss: $\mathcal{L}_{\text{ISCL}}$ from Eq. (5);

19      Total loss: $\mathcal{L} \leftarrow \mathcal{L}_{\text{reg}} + \mathcal{L}_{\text{gate}} + \lambda \cdot \mathcal{L}_{\text{ISCL}}$;

20      Update model parameters via backpropagation;

---

Table 7: Overview of the datasets.

| Dataset | #Samples | Unit | Range (Y) | Description |
|---------|----------|------|-----------|-------------|
| **Caco2** | 906 | logPapp | [-7.5, -3.6] | Caco-2 cell permeability; simulates intestinal absorption. |
| **PPBR** | 1,000 | % | [7.8, 100] | Plasma protein binding ratio; reflects drug availability in blood. |
| **Lipo** | 4,200 | logD | [-1.3, 4.5] | Lipophilicity; influences drug absorption and distribution. |
| **LD50** | 7,385 | mg/kg | [0.01, 7.1] | Median lethal dose; widely used measure of acute toxicity. |
| **QM9** | 133,885 | eV | [-14.2, 0.2] | HOMO–LUMO gap of small molecules; quantum chemistry dataset. |

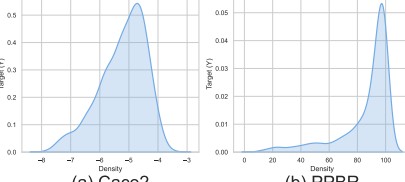
(a) Caco2

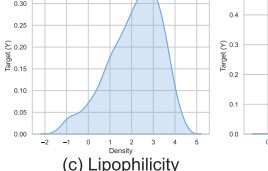
(b) PPBR

(c) Lipophilicity

(d) LD50

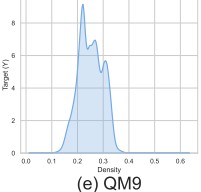
(e) QM9

Figure 9: Target distributions of the datasets.

## A.2 DATASET DETAILS

The four molecular property datasets used in this study vary in task type, scale, and target range, as summarized in Table 7. Caco2 and PPBR contain fewer than 1,000 samples, while Lipo and LD50 are substantially larger, with over 4,000 and 7,000 compounds respectively. The prediction tasks cover a diverse set of pharmacokinetic and toxicological endpoints.

Figure 9 illustrates the target distributions of each dataset. All four exhibit varying degrees of imbalance, with PPBR and LD50 showing long-tailed or skewed patterns, while Caco2 and Lipo have more compact but unevenly sampled target ranges. These distributional characteristics pose challenges for regression models, particularly in underrepresented regions of the target space.

## A.3 MODEL COMPONENTS AND IMPLEMENTATION DETAILS

We describe the architectural configurations of key modules used in our experiments, including the molecular encoders (backbones), regressor head, and expert networks within DistRouting.

### A.3.1 BACKBONE ARCHITECTURES

We evaluate DistRouting on three types of molecular encoders:

- **GAT:** A 4-layer Graph Attention Network with hidden size 512, ReLU activation, dropout rate 0.2, and Jumping Knowledge (JK) via concatenation. No normalization layers are used.

- **DeeperGCN:** A 4-layer graph convolutional network based on GENConv blocks. Each layer uses residual connections, PReLU activation, and batch normalization, with a hidden size of 512. Global mean pooling is used for readout.

- **ChemBERTa:** A transformer-based encoder for SMILES strings. We use the pretrained `DeepChem/ChemBERTa-77M-MLM` model with 6 transformer layers, 384 hidden dimensions, and 12 attention heads. The [CLS] token embedding serves as the molecular representation.

- **GROVER:** A graph-transformer pre-trained on large-scale molecular data. We use the `grover_base` checkpoint.

### A.3.2 REGRESSOR HEAD

The output representation from the DistRouting module is passed to a two-layer feedforward regressor with a ReLU activation in between. The regressor maps from the input feature dimension to a hidden dimension (512 in our experiments), and finally outputs a scalar property prediction.

### A.3.3 EXPERT NETWORKS

Each expert is a two-layer feedforward network. The input is first projected to a lower expert-specific hidden dimension, followed by ReLU activation and a second linear transformation back to the original size. Experts are initialized with Xavier initialization. Only the top-$k$ experts selected by the router contribute to each prediction.

### A.3.4 EXPERIMENTS COMPUTE RESOURCES

All experiments were conducted on a computing server equipped with an NVIDIA A100 GPU with 80GB memory, running Ubuntu 24.04.2 LTS. Each task was trained on a single GPU.

### A.3.5 HYPERPARAMETERS SETTINGS

Table 8 summarizes the dataset-specific hyperparameters used in DistRouting, including the ISCL loss weight $\lambda$ (Eq. equation 9) and the repulsion strength parameter $\gamma$ (Eq. equation 8), which controls the penalty for dissimilar sample pairs. These hyperparameters were selected via grid search over $\lambda \in \{0.1, 1.0\}$ and $\gamma \in \{2, 4, 10, 20, 50\}$.

Table 8: Hyperparameter settings.

| Dataset | $\lambda$ | $\gamma$ |
|---|---|---|
| Caco2 | 0.1 | 4 |
| PPBR | 0.1 | 4 |
| Lipophilicity | 1.0 | 10 |
| LD50 | 1.0 | 10 |

## A.4 ADDITIONAL RESULTS

### A.4.1 VALIDATION MAE CURVES

Figure 10 illustrates the MAE on the validation set across training epochs. We observe that DistRouting consistently converges to lower error compared to the vanilla encoders, and in some cases achieves faster convergence.

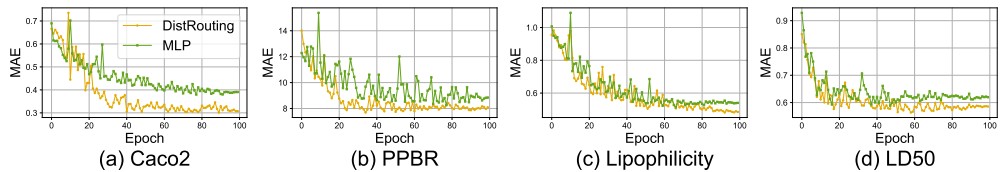

(a) Caco2      (b) PPBR      (c) Lipophilicity      (d) LD50

Figure 10: **MAE on the validation set over training epochs.** Across all four regression tasks, DistRouting (yellow) reaches lower error compared to the vanilla encoder (green).

### A.4.2 OVERALL PERFORMANCE

Table 9 reports PCC across the four datasets. DistRouting consistently improves correlation compared to the vanilla backbones, showing stronger alignment between predictions and targets.

Table 9: PCC ($\uparrow$) on the four datasets. Bold numbers indicate the best result in each column.

| Method | Caco2 | Lipophilicity | PPBR | LD50 |
|---|---|---|---|---|
| DeeperGCN | $0.788 \pm 0.010$ | $0.810 \pm 0.007$ | $0.552 \pm 0.021$ | $0.568 \pm 0.005$ |
| DeeperGCN + DistRouting | $\mathbf{0.832 \pm 0.014}$ | $\mathbf{0.826 \pm 0.007}$ | $\mathbf{0.623 \pm 0.009}$ | $\mathbf{0.607 \pm 0.024}$ |
| GAT | $0.794 \pm 0.022$ | $0.767 \pm 0.007$ | $0.618 \pm 0.013$ | $0.540 \pm 0.027$ |
| GAT + DistRouting | $\mathbf{0.819 \pm 0.015}$ | $\mathbf{0.794 \pm 0.004}$ | $\mathbf{0.624 \pm 0.013}$ | $\mathbf{0.613 \pm 0.042}$ |
| ChemBERTa | $0.778 \pm 0.018$ | $\mathbf{0.797 \pm 0.004}$ | $0.527 \pm 0.007$ | $0.545 \pm 0.013$ |
| ChemBERTa + DistRouting | $\mathbf{0.827 \pm 0.012}$ | $0.793 \pm 0.004$ | $\mathbf{0.627 \pm 0.014}$ | $\mathbf{0.611 \pm 0.020}$ |
| GROVER | $0.735 \pm 0.012$ | $0.839 \pm 0.012$ | $0.531 \pm 0.032$ | $0.503 \pm 0.070$ |
| GROVER + DistRouting | $\mathbf{0.792 \pm 0.010}$ | $\mathbf{0.870 \pm 0.004}$ | $\mathbf{0.600 \pm 0.022}$ | $\mathbf{0.690 \pm 0.026}$ |

### A.4.3 REGION-WISE MAE COMPARISON ACROSS BACKBONES

Figure 11 compares MAE across head, body, and tail regions for different backbones. DistRouting yields improvements across most regions and backbones, with particularly notable gains in the head and tail, while maintaining stable performance in the body region.

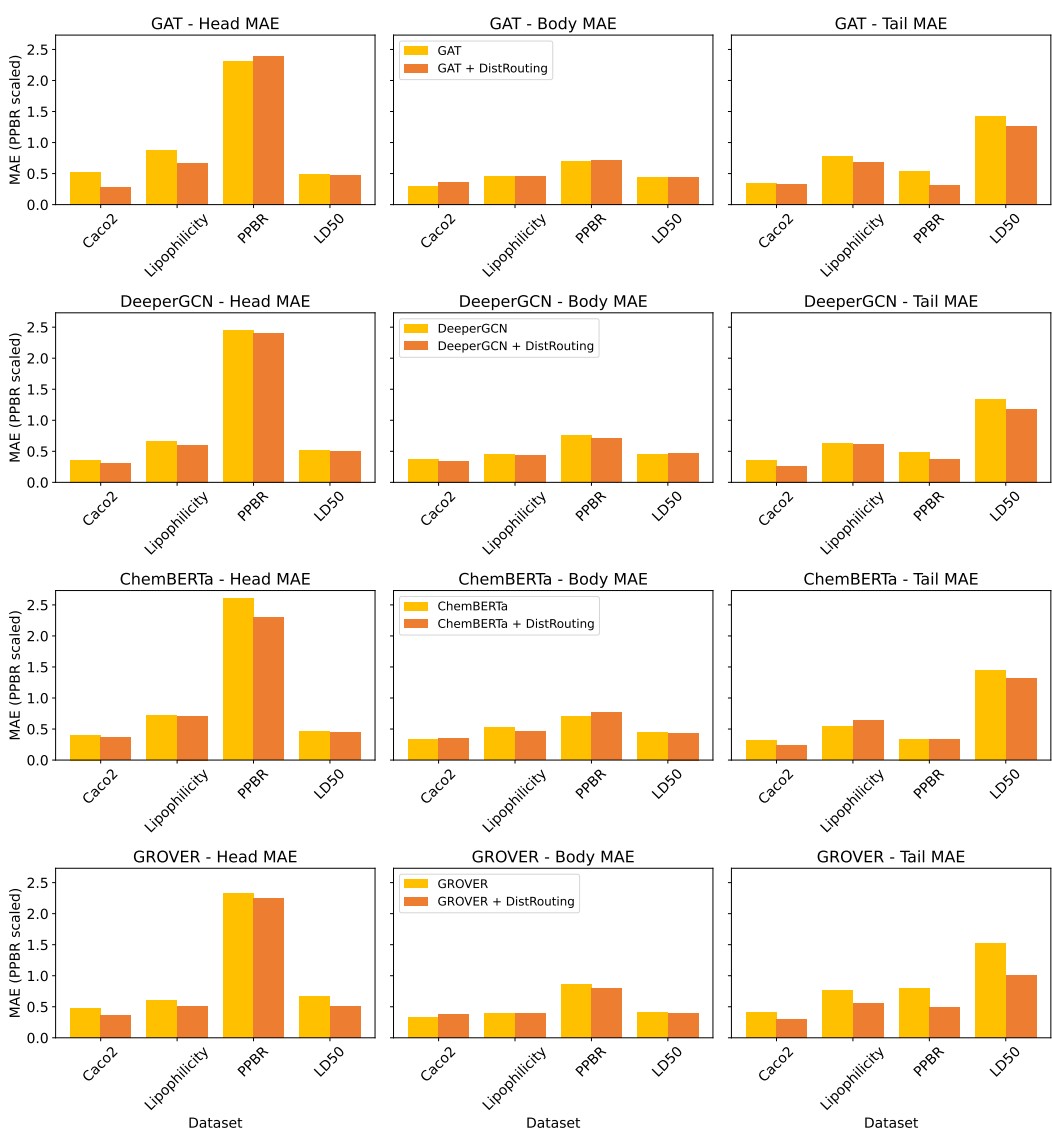

Figure 11: Region-wise MAE across four datasets and three backbone models. Red bars denote results with DistRouting, while green bars correspond to the baseline models without distribution-enhanced routing. Note that PPBR values have been scaled down to allow visual comparison.

### A.4.4 EMBEDDING–TARGET CORRELATION

We conducted additional analyses to directly assess the alignment between the learned embeddings and the target values. Specifically, we evaluated:

- Linear R², computed by fitting a linear regressor on the embeddings using 5-fold cross-validation

- Centered Kernel Alignment (CKA) similarity between the embeddings and target values

Table 10 reports the results, showing that ISCL substantially improves the correlation between embeddings and target values on both datasets.

Table 10: Alignment between embeddings and target values with and without ISCL.

| Dataset | Metric | w/ ISCL | w/o ISCL |
|---------|--------|---------|----------|
| LD50 | Linear R² (5-fold) ↑ | $0.7849 \pm 0.0279$ | $0.3894 \pm 0.1696$ |
| LD50 | CKA Similarity ↑ | 0.7029 | 0.3473 |
| Lipophilicity | Linear R² (5-fold) ↑ | $0.8810 \pm 0.0167$ | $0.5443 \pm 0.0428$ |
| Lipophilicity | CKA Similarity ↑ | 0.8615 | 0.5335 |

### A.4.5 GATING SUPERVISION

To better understand how gating behaves under distribution-aware supervision, we visualize the distribution of expert assignments across target values. Figure 12 shows representative results on Caco2, PPBR, and Lipophilicity.

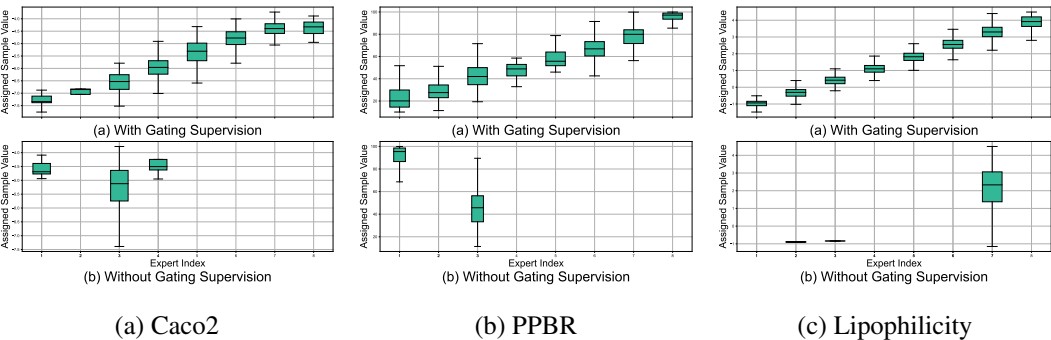

(a) Caco2      (b) PPBR      (c) Lipophilicity

Figure 12: Distribution of expert assignments across target values on the Caco2, PPBR and Lipophilicity dataset.

### A.4.6 EFFECT OF ISCL LOSS WEIGHT ON PERFORMANCE

Table 11 presents the impact of varying the ISCL loss weight $\lambda$ across four datasets. A smaller weight ($\lambda = 0.1$) generally yields the best results on Caco2 and PPBR, suggesting that a modest contrastive signal is sufficient to enhance representation learning in most settings. For Lipophilicity, performance remains relatively stable across different values of $\lambda$, indicating low sensitivity to the ISCL weight. In contrast, LD50 exhibits a different trend: performance is suboptimal at low weights, which is discussed in Section 4.4.

Table 11: Validation MAE under different ISCL weights $\lambda$.

| $\lambda$ | Caco2 | PPBR | Lipo | LD50 |
|-----------|-------|------|------|------|
| 0.1 | **0.315** | **7.849** | **0.508** | 0.643 |
| 1.0 | 0.326 | 8.427 | 0.509 | 0.616 |
| 2.0 | 0.345 | 8.403 | 0.509 | 0.624 |
| 3.0 | 0.377 | 8.267 | 0.511 | **0.605** |

### A.4.7 QM9 PERFORMANCE

To evaluate scalability, we further tested DistRouting on the large-scale QM9 dataset. Table 12 shows that incorporating DistRouting into UniMol leads to clear overall improvements in both MAE and PCC. Region-wise analysis on the HOMO–LUMO gap (Table 13) further confirms consistent gains across head, body, and tail regions.

Table 12: Performance on QM9. MAE ($\downarrow$) and PCC ($\uparrow$) are reported. Bold numbers indicate the best result.

| Method | MAE $\downarrow$ | PCC $\uparrow$ |
|---|---|---|
| UniMol-MLP | $0.0084 \pm 0.0000$ | $0.9690 \pm 0.0003$ |
| UniMol + DistRouting | $\mathbf{0.0066 \pm 0.0001}$ | $\mathbf{0.9790 \pm 0.0005}$ |

Table 13: Region-wise MAE ($\downarrow$) on QM9 HOMO–LUMO gap. Best between each pair is bolded.

| Method | Head MAE $\downarrow$ | Body MAE $\downarrow$ | Tail MAE $\downarrow$ |
|---|---|---|---|
| UniMol-MLP | $0.0109 \pm 0.0001$ | $0.0079 \pm 0.0000$ | $0.0074 \pm 0.0001$ |
| UniMol-DistRouting | $\mathbf{0.0086 \pm 0.0001}$ | $\mathbf{0.0062 \pm 0.0001}$ | $\mathbf{0.0059 \pm 0.0000}$ |

## A.5 DISTRIBUTION SIMILARITY EVALUATION VIA JS DISTANCE

To evaluate the similarity between the predicted and true target distributions, we compute the Jensen–Shannon (JS) distance, which is the square root of the Jensen–Shannon divergence—a symmetric and smoothed variant of the Kullback–Leibler (KL) divergence. Given two probability distributions $P$ and $Q$ over the same discrete support, the JS distance is defined as:

$$\text{JSD}(P \parallel Q) = \sqrt{\frac{1}{2}D_{\text{KL}}(P \parallel M) + \frac{1}{2}D_{\text{KL}}(Q \parallel M)} \quad \text{where} \quad M = \frac{1}{2}(P + Q)$$

Here, $\text{KL}(\cdot \parallel \cdot)$ denotes the Kullback–Leibler divergence. A smaller JS distance indicates greater similarity between the two distributions, with a value of zero signifying identical distributions.

## A.6 USE OF LARGE LANGUAGE MODELS

We acknowledge the use of a large language model (LLM) as a writing assistant in preparing this manuscript. The LLM was used solely to improve clarity, conciseness, and readability, as well as to suggest refinements in narrative flow. All scientific ideas, methods, experiments, and analyses are entirely the work of the authors.

