# OpenReview forum: "Distribution-Guided Expert Routing for Imbalanced Molecular Property Regression"
_ICLR.cc/2026/Conference — ICLR 2026 Conference Withdrawn Submission_

### Official Review · Reviewer_JkJF · 2025-10-31

**Soundness:** 3
**Presentation:** 3
**Contribution:** 2
**Rating:** 4
**Confidence:** 4

**Summary:**

This paper proposes DistRouting, a distribution-guided expert routing framework for imbalanced molecular property regression that uses mixture-of-experts architecture to route molecules to specialized experts based on predicted target ranges.

**Strengths:**

The paper addresses an important practical problem of target distribution imbalance in molecular property prediction, which is relevant for real-world applications in drug discovery.

**Weaknesses:**

1. The paper's innovation is limited as it essentially applies existing Mixture of Experts (MoE) architecture to molecular property prediction. The authors fail to provide theoretical analysis or proof of why this specific expert routing approach is superior to alternatives. There is no theoretical guarantee about expert specialization or load balancing.

2. The paper claims that 6-16 experts perform well, but it remains unclear what each expert's responsibility is and what they actually learn. Critical questions are left unanswered:
- Why is k=2 chosen for top-k routing?
- Most importantly, I am curious about expert selection for each molecule: Are all expert scores similar? Do the same experts get selected repeatedly? It seems highly possible that the same experts got selected, is there any strategy to handle such cases?
- The authors should provide detailed analysis of expert utilization patterns and routing behavior, but this analysis is missing.

3. The dataset evaluation is insufficient:

- Limited TDC coverage: TDC has many regression tasks available, yet the authors only select 4 datasets without proper justification.
- Missing standard benchmarks: The paper fails to evaluate on important MoleculeNet benchmarks including ESOL, FreeSolv, and Lipophilicity, which are widely used in the community.

4. It is unclear why the authors do not compare against recent state-of-the-art methods, including:
BioT5: "BioT5: Enriching Cross-modal Integration in Biology with Chemical Knowledge and Natural Language Associations"
BioT5+: Enhanced version of BioT5
MolXPT: "MolXPT: Wrapping Molecules with Text for Generative Pre-training"
MolFormer: "Large-scale chemical language representations capture molecular structure and properties"
Atomas: "Atomas: Hierarchical Alignment on Molecule-Text for Unified Molecule Understanding and Generation"

**Questions:**

See weaknesses.

---

### Official Review · Reviewer_jUvy · 2025-11-01

**Soundness:** 2
**Presentation:** 2
**Contribution:** 2
**Rating:** 2
**Confidence:** 4

**Summary:**

This paper proposes DistRouting, a mixture-of-experts framework for molecular property regression under imbalanced target distributions. The method partitions the continuous target space into intervals and assigns specialized experts to each region. The routing mechanism combines molecular embeddings with RDKit physicochemical descriptors, supervised by a KL divergence loss based on soft interval labels. The authors also introduce an interval-aware supervised contrastive learning (ISCL) loss to structure the representation space. Experiments on four molecular property datasets show consistent improvements over baseline encoders, particularly in rare target regions.

**Strengths:**

The paper targets a well-motivated problem in molecular property prediction, which is target distribution imbalance, where models tend to overfit to dense regions while underperforming on rare but critical ones. The observation that structurally dissimilar molecules can have similar properties (Figure 1) provides good motivation for why existing reweighting/resampling methods may fail.
The framework is simple and general. DistRouting is designed as a plug-in module that can be integrated with diverse molecular encoders (GNNs, transformers, 3D models). The consistent improvements across four different backbones demonstrate good generalizability.
The region-wise analysis (Table 2) shows substantial improvements in head and tail regions across most datasets and backbones (30 out of 32 cases), which directly validates the effectiveness in addressing imbalance.

**Weaknesses:**

1. Missing related work on few-shot and contrastive learning for molecular imbalance: The paper employs contrastive learning to handle data imbalance but fails to cite or compare with recent work that combines few-shot learning and contrastive learning for similar molecular property prediction challenges:

 - MolFeSCue (Zhang et al., "MolFeSCue: enhancing molecular property prediction in data-limited and imbalanced contexts using few-shot and contrastive learning," Bioinformatics, 2024, https://academic.oup.com/bioinformatics/article/40/4/btae118/7616990) explicitly addresses imbalanced molecular regression by combining few-shot meta-learning with supervised contrastive learning, using a dynamic contrastive loss function to handle class imbalance.

 - Meta-MGNN (Guo et al., "Few-shot graph learning for molecular property prediction," WWW, 2021, https://dl.acm.org/doi/abs/10.1145/3442381.3450112) applies molecular graph neural networks with a meta-learning framework for optimization, incorporating self-supervised modules to exploit unlabeled molecular information.

 - PAR (Wang et al., "Property-aware relation networks for few-shot molecular property prediction," NeurIPS, 2021, https://proceedings.neurips.cc/paper/2021/hash/91bc333f6967019ac47b49ca0f2fa757-Abstract.html) introduces property-aware embedding functions and adaptive relation graph learning to handle few-shot molecular property prediction.

2. The connection and distinction between expert routing for imbalance and few-shot learning paradigms deserve explicit discussion, as both aim to improve performance on underrepresented samples.
Insufficient justification for design choices:

 - Why uniform interval partitioning? The paper mentions in limitations that "property-aware partitioning" could be explored, but provides no justification for why equal-width intervals are chosen initially. Given that target distributions are highly non-uniform (Figure 9), adaptive partitioning based on data density seems more principled.
 - Why combine embeddings and RDKit features additively (Eq. 1)? No ablation compares alternative fusion strategies (concatenation, gating, etc.). The additive combination assumes equal importance, which may not hold.
 - Top-k=2 seems arbitrary: While Section 5 shows sensitivity to the number of experts, there's no analysis of how top-k should scale with the number of experts or dataset size.


3. Inconsistent benefits of ISCL: Table 3 and discussion in Section 4.4 reveal that ISCL actually hurts performance on LD50 at the default weight, and the paper resorts to post-hoc weight tuning (Table 11) to recover performance. This raises concerns:

 - Why does ISCL behave so differently across datasets?
The claim that "ISCL meaningfully enhances the structure of representation space" (Section 4.4) contradicts the regression performance on LD50.
 - Table 10 shows improved embedding-target alignment, but this doesn't translate to better predictions consistently, suggesting the alignment metrics may not be the right objective.

4. Limited novelty in contrastive learning component: The ISCL formulation (Eq. 7-8) is a straightforward adaptation of supervised contrastive learning with soft labels based on Gaussian kernels. The contribution here is incremental.

**Questions:**

1. Can you explain why few-shot learning techniques are not applicable or beneficial for your setting?
Have you experimented with adaptive or data-driven interval partitioning strategies? How sensitive is performance to the choice of intervals?
2. Can you explain why ISCL has such inconsistent effects across datasets? Is there a principled way to set the weight λ without dataset-specific tuning?
3. Why do existing imbalance methods (DenseWeight, FDS, LDS) perform so poorly on your datasets? Have you verified your implementations against the original papers?
4. What is the computational overhead of DistRouting compared to baseline models? Can you provide training time and memory comparisons?

---

### Official Review · Reviewer_NHP9 · 2025-11-01

**Soundness:** 1
**Presentation:** 2
**Contribution:** 3
**Rating:** 2
**Confidence:** 3

**Summary:**

This paper introduces DistRouting, a framework designed to tackle imbalanced regression for molecular property prediction. The core idea is to use a Mixture-of-Experts (MoE) architecture where different experts specialize in distinct intervals of the continuous target property distribution. The framework routes input molecules to the appropriate experts using a gating mechanism guided by both learned molecular embeddings and pre-computed RDKit descriptors. Additionally, an interval-aware supervised contrastive loss is proposed to structure the embedding space. The method is presented as a plug-in module and evaluated across several molecular encoders on a number of benchmark datasets.

**Strengths:**

- The core idea of a distribution-aware Mixture-of-Experts, where experts specialize on different regions of the target distribution, is an intuitive and interesting approach to the problem. To the best of my knowledge, this is a novel contribution in the molecular property prediction domain.
- The authors demonstrate the versatility of their method by applying it as a plug-in module to four different molecular encoder architectures (GAT, DeeperGCN, ChemBERTa, GROVER), showing consistent improvements over the respective baselines.
- The paper addresses the problem of imbalanced regression in molecular property prediction, which is a practical challenge in the field.

**Weaknesses:**

- The experimental setup does not adequately control for confounding variables, making it difficult to attribute performance gains solely to the proposed routing mechanism:
    - The DistRouting-enhanced models seemingly have more parameters than the baselines. The authors attempt to address this in Table 5 by comparing different model MLP sizes, but this seems like an after-the-fact analysis rather than a controlled experiment. A more rigorous approach would have been to match the parameter counts from the outset.
    - The apparently extreme levels of sensitivity to MLP size in Table 5 raise concerns about the robustness of the findings—perhaps the evaluation datasets are too small to draw reliable conclusions, or the models are not well-regularized.
    - It is unclear if the baseline models are also provided with the RDKit descriptors. If not (which is what the current text suggests), the comparison is unfair, as the improvements may simply come from using these additional informative features rather than the novelty of the routing architecture.
- The evaluation is conducted on datasets that are extremely small by modern machine learning standards:
    - The paper refers to QM9 as a "large" dataset, which is misleading. While QM9 is larger than the other datasets used, it is still relatively small (around 130k samples) compared to contemporary benchmarks in molecular machine learning.
    - QM9 evaluation, which is the only dataset approaching a moderate size, is limited to a single property (HL gap), whereas the benchmark contains 12, and the table is pushed to the appendix.
    - There is a lack of comparison to any established state-of-the-art (SOTA) methods on standard benchmarks. This makes it difficult to gauge the practical significance of the results. Below are some suggestions for more meaningful evaluations:
        - For small to medium-sized small molecule datasets, the MoleculeNet benchmark suite is widely used (and would provide a better ability to compare against SOTA). This covers both 2D and 3D molecular property prediction tasks.
        - For larger molecular datasets, PCQM4Mv2 (from OGB-LSC) and OMol25 are good candidates.
        - For large-scale datasets outside of small molecules, the Open Catalyst 2020 (OC20) and Open Catalyst 2022 (OC22) datasets are relevant for materials and catalysis applications.
- Presentation and clarity issues:
    - Nearly all tables lack units for the reported metrics, and figures are not always self-contained (e.g., the meaning of colors in Figure 4 is not explained).
    - The ablation results are reported without error bars, which is problematic given the small dataset sizes and the variability that is common in such settings.

Based on the current evaluation, the experimental design is not rigorous enough to convincingly demonstrate that the proposed method's architectural novelty is the primary driver of the observed improvements. The gains could plausibly be explained by the use of more parameters and additional features (RDKit descriptors) that are not available to the baselines. While the idea is interesting, the lack of strong, controlled experiments and comparisons to SOTA methods on larger-scale benchmarks limits the impact of the contribution.

**Questions:**

1. One core argument of this paper is that locality violation (i.e., structurally dissimilar molecules having similar properties) is a significant challenge for molecular representation learning. The paper argues that feature calibration methods risk "blurring critical distinctions by forcing structurally distinct yet label-similar molecules closer in the representation space." However, doesn't the proposed Interval-Aware Supervised Contrastive Learning (ISCL) appear to do something very similar by pulling samples within the same target interval closer together? Could you clarify the distinction and reconcile this apparent contradiction in motivation? Does this imply that the assumption of locality violation is not as problematic as initially suggested?
2. For the main results in Table 1, were the baseline models (DeeperGCN, GAT, etc.) also given access to the RDKit descriptors used by DistRouting? If not (which is what the current text suggests), how can we be sure the improvements are not just from the addition of these features?
    - The current text attempts to address this in the ablation study (Table 3), but, as with the parameter count issue, this seems backwards. This should have been controlled for from the start. The addition of these features seems completely orthogonal to the architectural novelty of the routing mechanism.
    - Can the authors also justify their claim that "RDKit guidance causes moderate drops but still outperforms the baseline encoder, indicating that the routing mechanism provides useful inductive bias"? In 3/4 of the cases, the delta between the full DistRouting and the "w/o RDKit Guidance" ablation is larger than the delta between the "w/o RDKit Guidance" and the vanilla baseline.
3. Could the authors comment on how their method differs from some similar distribution regression learning methods, such as "Distribution Learning for Molecular Regression" (arXiv:2407.20475) and "Improving Regression Performance with Distributional Losses" (arXiv:1806.04613)? I understand these works have different focuses, but it would be helpful for the authors to position their contributions in relation to these existing methods.
4. Can you explain what the colors represent in the t-SNE plots of Figure 4? Further, can you elaborate on how these visualizations "suggest that ISCL meaningfully enhances the structure of the representation space"?

**Minor Comments and Nits**

- In the ablation study (Table 3), it would be helpful to include the performance of the vanilla baseline encoder for easier comparison, especially since the text makes direct comparisons to it.
- The appendix refers to "three backbone models" in a few places (lines 683 and 798), despite four being discussed throughout the paper.

---

### Official Review · Reviewer_jeQC · 2025-11-02

**Soundness:** 3
**Presentation:** 3
**Contribution:** 3
**Rating:** 4
**Confidence:** 4

**Summary:**

The paper presents DistRouting, MoE architecture that dynamically routes molecular samples to experts based on their property ranges. Each expert is
designated to a specific property range. The routing is guided by physiochemical descriptors obtained from RDKit. Additionally, ISCL is proposed, a
contrastive type loss, that distinguishes samples from different intervals. The method is tested on multilpe benchmarks.

**Strengths:**

- The paper introduces an architecture for handling label imbalance via distribution gudied expert routing.The main contribution there is the MoE
  style architecture which is guided by target distribution.

- Chemical priors, obtained from RDKit, are incorporated into the routing decision.

- An interval aware contrastive loss in proposed and is tailored specifically for regression.

**Weaknesses:**

- The method is crafted specifically for continuous-valued modeluclar property prediction tasks and may not necessarily work on molecular learning
  tasks such as classification. This is significant given that in real scenarios, one is normally confronted with a combination of both discrete and
  continuous properties.

- The model realies on a combination of learned and fixed features (those obtained from RDKit) making it have a strong reliance on such features. As
  shown in the ablation, removing these features has a strong impact on the resulting model performance.

**Questions:**

- Given that DistRouting relies on intervals, how is the method sensitive to the number/scope of these intervals? What happens when intervals are
   adaptive?

 - Can DistRouting be adapted for classification tasks or multi-task molecular prediction tasks? What architectural choices have to be made to
   accomodate these? Or even in multi-property regression tasks?

 - What happens when the RDKit features are not available? If these features were not available, can a surrogate be used? Also are all 200 descriptors
   necessary?

---

### Note · Authors · 2026-01-11

I have read and agree with the venue's withdrawal policy on behalf of myself and my co-authors.